# Spatiotemporal Influences of LULC Changes on Land Surface Temperature in Rapid Urbanization Area by Using Landsat-TM and TIRS Images

Eduardo Andre Kaiser [1,*,†], Silvia Beatriz Alves Rolim [1,†], Atilio Efrain Bica Grondona [1,†], Cristiano Lima Hackmann [1,†], Rodrigo de Marsillac Linn [2,†], Pâmela Suélen Käfer [1,†], Nájila Souza da Rocha [1,†] and Lucas Ribeiro Diaz [1,3,†]

1 State Research Center for Remote Sensing and Meteorology (CEPSRM), Federal University of Rio Grande do Sul (UFRGS), Porto Alegre 90040-060, RS, Brazil; silvia.rolim@ufrgs.br (S.B.A.R.); atilio.grondona@ufrgs.br (A.E.B.G.); cristiano.hackmann@ufrgs.br (C.L.H.); pamela.kafer@ufrgs.br (P.S.K.); najila.rocha@ufrgs.br (N.S.d.R.); l.ribeiro.diaz@vu.nl (L.R.D.)
2 Coordination of Evaluation of Public Policies and Results, Prefeitura Municipal de Porto Alegre, Porto Alegre 90010-170, RS, Brazil; rodrigoml@portoalegre.rs.gov.br
3 Faculty of Science, Vrije Universiteit Amsterdam, 1081 HV Amsterdam, The Netherlands
* Correspondence: eduardo.kaiser@ufrgs.br; Tel.: +55-55-996961633
† These authors contributed equally to this work.

**Abstract:** The inverse correlation between NDVI and LST is widely known for its long time series. However, when more specific statistical tests were performed, subtle differences in the correlation behavior over time are more clearly observed. In this work, regression analyses were performed between NDVI and LST at intervals of approximately 10 years, quantifying this relationship for an area of transition from vegetation to urban occupation from 1985 to 2018. The removal of vegetation cover (reduction of 51% to 7% in grassland and 14.4% to 0.6% in forest) to occupy impermeable surfaces ( increase of 31% to 91% in urban areas) caused an average LST increase of 4.18 °C when compared to the first and last decades of the historical series. Temporal analysis allowed us to verify the increase in temperature in the four seasons. The largest difference was 6.36 °C between the first and last decade of autumn, 4.40 °C in spring, 4.09 °C in summer, and 2.41 °C in winter. The results also show that LST has a negative correlation with NDVI, especially in urban areas, with an increase in this correlation during the period (1989: R = −0.55; 1999: R = −0.58; 2008: R = −0.59; 2018: R = −0.76). Our study results will help policymakers understand the dynamics of temperature increases by adding scientifically relevant information on the sustainable organization of the urban environment.

**Keywords:** vegetation and urban modeling; land use and land cover change; land surface processes; Porto Alegre four districts

## 1. Introduction

In view of the urbanization process, which has intensified since the 18th century and is mainly associated with the occurrence of the rural exodus and the industrialization process, it is possible to verify the unrestrained growth of the population and urban spots in the global scenario. According to the World Urbanization Prospects report, produced by the United Nations Division, on a global scale, the urban population has been larger than the rural population since 2007 [1]. In Brazil, according to data from the census conducted by the Brazilian Institute of Geography and Statistics (IBGE) in 2010, 84.4% (169.9 million) of the population lived in an urban area [2].

The expansion of cities severely alters the natural biophysical environment through changes in land use and land cover (LULC). The removal of vegetation covers to develop urban activities causes changes not only in the hydrological cycle but mainly in energy balance, with regard to energy storage and transfer that occurred naturally before anthropic

interference [3,4]. As a consequence, there are changes in air temperature and humidity, wind speed, and direction. In this manner, the city starts to develop its own urban climate determined by these changes, by the regional climate, and by the local physical environment, configuring a Superficial Urban Heat Island (SUHI) [5,6]. According to Zhang and Sun [7], as the main urbanization reflection, there is an increase in the Earth's surface temperature (LST) and the consequent formation of SUHI, also known as heat core, thermal or humid cores, heated core, or even as heat pockets. This increase occurs because of anthropic interferences occuring in the environment by replacing natural surface covers with materials with high heating capacity. In addition, there is a decrease in urban green areas that would be responsible for reducing LST through the evapotranspiration process [8,9].

Vegetation presence influences LST through the absorption and selective reflection of solar radiation, regulating the exchange of latent and sensitive heat [10–12]. In an urban environment, the correlation between this presence and lower temperature areas is established. Such correlations can be observed by using generalized definitions of surface greenness or by using the Normalized Difference Vegetation Index (NDVI) values detected using remote sensing techniques [13–16].

In the last decades, some studies were conducted regarding the relationship between LULC and LST variables in large cities such as Tokyo [17], Bangkok [9] and Inner Hanoi [18] and with low spatial resolution [13,15,19,20]. The impact of urbanization combined with changes in LULC and seasonal effects on LST and SUHI intensity was also verified in seven major urban districts (Barisal, Chattogram, Dhaka, Mymensingh, Rajshahi, Rangpur, and Sylhet) of Bangladesh [21] and in cities such as Raipur [22], Chittagong [23], Shiraz [24], Wuhan [25], Bengaluru [26], Calcutá [27],and Tehran [28]. Furthermore, most of these studies related variables according to specific conditions in the areas of study. However, the LST, NDVI, and LULC relationship occurs differently over time and according to land use and cover in a large metropolitan area compared to a medium-sized urban center.

The inverse correlation between NDVI and LST is widely known for long time series [22,24,29–31]. However, this approach does not always work, because some changes can occur at specific periods without being registered. This subtle anomalous behavior is enhanced and characterized in this study, by using appropriated statistical performance. Regression analyses were used in NDVI and LST data at adjusted intervals of approximately 10 years. The results allowed quantifying a different pattern between NDVI and LST for an area of transition from vegetation to urban occupation from 1985 to 2018. This unnoticed behavior can help to better understand the dynamics of temperature increases, adding scientifically relevant information on the sustainable organization of urban environments.

From the beginning of the 20th century to the present, the municipality of Porto Alegre-Rio Grande do Sul, Brazil, has experienced accelerated urban growth mainly due to the industrial and labor occupation process becoming an economic and population center at the regional level. The urban spot rapid transformation and expansion caused not only central and rural area densification but also urban conurbation phenomenon. According to Frumkin [32], this phenomenon is known as the urban sprawl and human activities diversification, demanding enormous planning and management capacity from the municipalities governments and technical staff. The city is among the ten smartest and most connected cities in Brazil (ninth position), being the 4th in entrepreneurship, 6th in technology and innovation, 13th in economy, and 15th in health. Although most large urban centers have agencies responsible for urban planning and the maintenance of public services, many lack space-time data and information that serve as a basis for decision making.

In this sense, the LST evolution diagnosis and characterization have been shown to be a potential instrument for urban space management [33]. However, the lack of detail regarding the LST and LULC spatial and temporal variation in urban environments ends up harming mitigation actions by the government. Once data on the spatial differences between the intraurban and rural temperatures are acquired, it is possible to measure and mitigate possible SUHI formation. Thus, urban planners and designers are allowed to

suggest measures to adjust LST and the associated effects of SUHI from the management of LULC composition [18]. LST, NDVI, and LULC spatio-temporal variations can be dimensioned using remote sensing techniques. Several works on the urban surfaces' thermal characterization from data obtained by a sensor can be observed in the world literature [34–39]. In Brazil, it is worth mentioning the work of [40–43]. The advent of these techniques made it possible to study SUHI, both on a local and global scale, since it allows the transformation of data in the thermal infrared spectral range for apparent surface temperature.

The objective of this work is to analyze the evolution of surface temperature between vegetated and urbanized areas, correlating 30 years of calculated data from LST and NDVI and identifying possible seasonal influences by uing regression analysis. The study shows the relationship between LST and NDVI during a rapid urbanization process and how changes in land use and land cover can affect this relationship.

## 2. Materials and Methods

The area of this study comprises the current urban location belonging to four districts located in Porto Alegre city, Rio Grande do Sul state (Brazil: Hípica, Campo Novo, Aberta dos Morros, and Restinga), as shown in Figure 1. This spot covers an area of 534.61 hectares, withan approximate population of 120,000 inhabitants (IBGE, 2010), and is located between the coordinates 51°15′–51°20′ W and 30°14′–30°18′ S. This area was defined by considering the changes in LULC due to the rapid urbanization process that occurred between the period of 1985 and 2018. As exposed by [36,44,45], this acceleration of urban growth resulted in significant changes in LULC and is responsible for increasing density and height-built areas.

The relief of Porto Alegre is characterized by a region of contact between the Planalto Uruguaio Sul-Rio-Grandense and the Lowland and/or Terras Baixas Costeiras, in addition to the sediments from the Peripheral Depression [46]. In altimetric terms, the municipality has altitudes ranging from 0.1 m on Ilha das Flores to 311.20 at its peak, Morro Santana. The study area has altitudes ranging from 14 m in the Hípica neighborhood (far west of the study area) to 21 m in the Restinga neighborhood (far east of the study area).

In turn, Porto Alegre's climate, according to the Koeppen classification, corresponds to the subtype "Cfa", for which its mean annual temperature results in 19.5 °C, an annual rainfall of 1300 mm [47], and an evapotranspiration annual mean of 937.38 mm [48].

The methodological procedures took place in five stages: 1. Definition of the study area; 2. LULC classification based on images from the Landsat 5 satellite TM sensor and the Landsat 8 satellite OLI sensor; 3. Calculation of LST from the TM sensor band 6 and the OLI sensor band 10; 4. LST temporal and spatial evolution analysis over the historical series; and 5. Spatio-temporal relationship between LST and LULC classes. The flowchart in Figure 2 shows each of the steps performed to construct the present study.

The choice of the period from 1985 to 2018 is based two reasons: 1. The history of the study area districts is associated with the appropriation of space from the creation of popular subdivisions and industrial parks. The Hípica district, which covers most of the study area, was officially created in 1991, the Restinga neighborhood (eastern region of the study area) was created in 1990, and the Campo Novo district was created in 2011. All of these areas have two characteristics in common: accelerated urban growth and consequent loss of the previously predominant rural landscape. Thus, they coincide with the history of the districts and the period used in the present study, and 2. the ideal period of a climate modeling study should cover series with at least 30 years of data. Faced with the difficulties associated with the availability of images with clear sky conditions, a longer time series allows the acquisition of a greater number of images in order to mainly cover seasonal variations of LST, NDVI, and LULC.

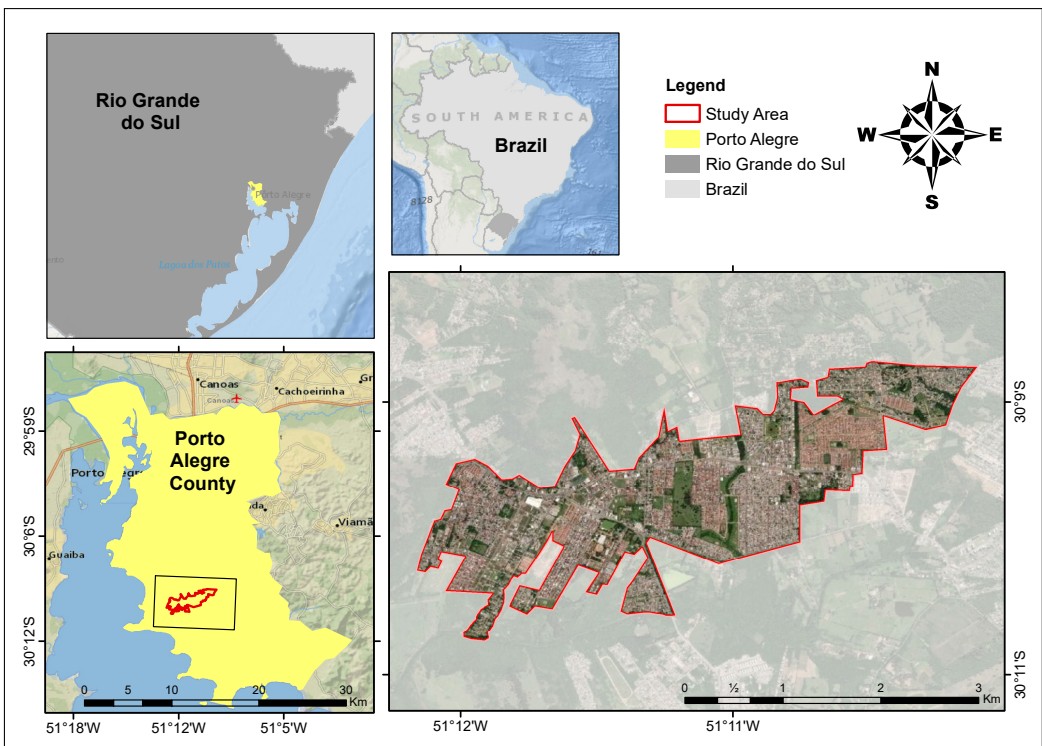

**Figure 1.** Location map of the urban spot of Hípica, Campo Novo, Aberta dos Morros, and Restinga neighborhoods in the municipality of Porto Alegre, Rio Grande do Sul, Brazil.

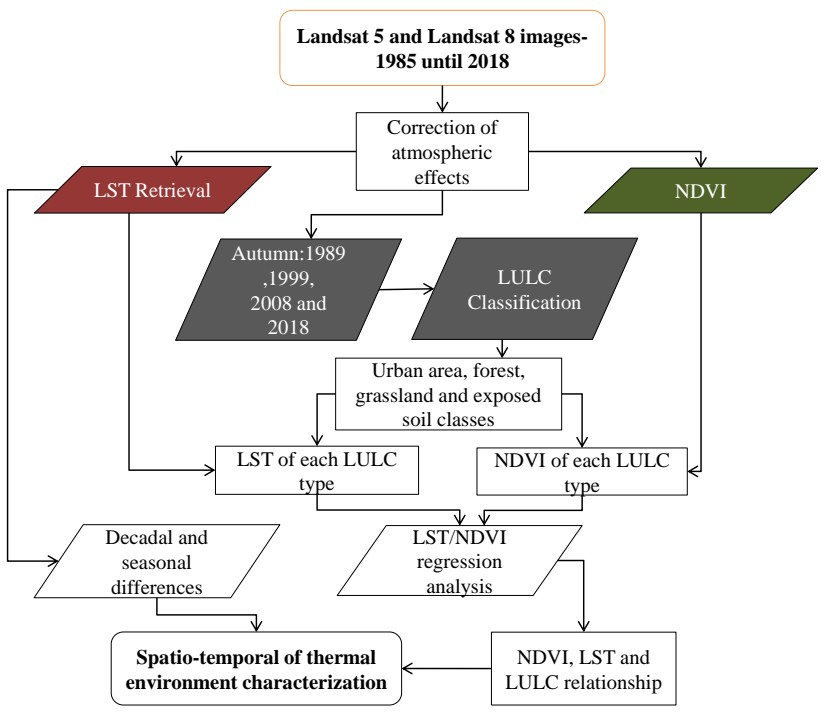

**Figure 2.** Flowchart of the methodological procedures used in the present study.

NDVI and thermal infrared images from the Landsat8 satellite Operational Terra Imager (OLI) and Thermal Infrared Sensor (TIRS) sensors were used to extract information about LULC and LST, respectively. The specific objectives are as follows: (1) derive LST from Landsat 5 and 8 satellites Thematic Mapper (TM) and TIRS sensors from 1985 to 2018;

(2) examine the LULC and LST spatial pattern and temporal variation for the same period; and (3) investigate the relationship between LST and LULC from NDVI.

Images from the TM- Landsat 5 sensor (30 m) and the OLI- Landsat 8 sensor (30 m) referring to four dates of the 34-year historical series studied in the present work were used to perform LULC classification: 29 September 1989, 25 October 1999, 3 October 2008, and 16 November 2018. The definition of dates took into account the approximate interval of 10 years, availability of images without the presence of clouds, and the associated seasonality; in this case, it is the spring season. The classification supervised by Maximum Likelihood was carried out [49]. Such a classifier qualifies as one of the most commonly used parametric algorithms for image classification, mainly due to its robustness [49–51]. The LULC classes used comprised those that are visually verified on the study area, which are named as follows: Forest, Urban Area, Grassland, and Exposed Soil.

Currently, different algorithms are used to retrieve LST from data obtained by orbital sensors, highlighting the mono-window algorithm [52], single-channel algorithm [53], radiation conduction equation [54], and split window algorithm [55]. The calculation of the surface temperature (LST) was performed as described by [56] using a single-channel algorithm. According to the authors, the first step consists in converting the digital number (DN) of each pixel in the image into spectral radiance and converting it to brightness temperature using the Planck function.

Since Landsat 8 and 5 satellites TIRS and TM sensors bands 10 and 6, respectively, have similar spectral ranges, several studies have carried out the use of these bands to retrieve LST [53,57–59] against TIRS sensor band 11 contamination by thermal energy outside the normal field of view (stray light effect) [60], which is also corroborated the results of indirect calibration [61].

Following the procedures, the present study retrieved superficial emissivity from NDVI (Equation (1)) [62] for the 117 images obtained:

$$\text{NDVI} = \frac{\rho_{\text{nir}} - \rho_{\text{red}}}{\rho_{\text{nir}} + \rho_{\text{red}}} \tag{1}$$

where NDVI is the Normalized Difference Vegetation Index, `nir` is the Near Infrared region, and `red` is the Red region. After obtaining NDVI values for each pixel, emissivity was estimated from the four cases shown in Table 1.

**Table 1.** Relationship between the Normalized Difference Vegetation Index and the Emissivity proposed by [62].

| NDVI | Land Surface Emissivity ($\epsilon_i$) |
|---|---|
| $\text{NDVI} < -0.185$ | 0.995 |
| $-0.185 \leq \text{NDVI} < 0.157$ | 0.970 |
| $0.157 \leq \text{NDVI} \leq 0.727$ | $1.0094 + 0.047\ln(\text{NDVI})$ |
| $\text{NDVI} > 0.727$ | 0.990 |

LST retrieval presents some complexity due to the surface not having characteristics similar to a blackbody in terms of thermal emission capacity. Moreover, the atmosphere and soil effects must be considered [63,64]. Therefore, to retrieve LST, Radiative Transfer Equation (RTE) inversion (Equation (2)) is applied to a given sensor channel and wavelength range:

$$L(sensor, \lambda) = [\epsilon_\lambda B_\lambda(T_s) + (1 - \epsilon_\lambda)L_{atm,\lambda}^{\downarrow}]\tau_\lambda + L_{atm,\lambda}^{\uparrow} \tag{2}$$

where *Lsensor* is the radiance measured by the sensor in $W/m^{-2}\, m^{-1}\, sr^{-1}$, $\epsilon_\lambda$ is the land surface emissivity (LSE), $B_\lambda(T_s)$ is the Planck's function given by Equation (3), $L^{\downarrow}$ is the

descending atmospheric radiation in $W/m^{-2}\,m^{-1}\,sr^{-1}$, $L^{\uparrow}$ is the ascending atmospheric radiation in $W/m^{-2}\,m^{-1}\,sr^{-1}$, and $T$ is the atmospheric transmittance:

$$B_\lambda(T_s) = \frac{C1\lambda^{-5}}{(exp(C2/\lambda T) - 1)} \tag{3}$$

where $C1$ and $C2$ are Planck radiation constants, with values of (each type of sensor has a value).

The parameters include the following: Descending and ascending atmospheric radiation and transmittance used in Equation (2) can be accessed through a website of the National Aeronautics and Space Administration (NASA, Washington, DC, USA) (http://atmcorr.gsfc.nasa.gov, acessed on 15 August 2021), where image information (such as sensor passing time, latitude and longitude, and season) is inserted to calculate these parameters. These parameters are available on the website from the year 2000, between 1985 and 1999, and the values of $L^{\downarrow}$, $L^{\uparrow}$, and $T$ generated by Moderate Spectral Resolution Atmospheric Transmittance Algorithm and Computer Model (MODTRAN) 4.0 v3r1 version 1.2 were used.

Since the relationship between vegetation areas and urban temperatures varies across space [65,66], for a comprehensive view of the impact caused by rapid urbanization on the LST, the LST means were calculated for each land use and land cover class, obeying the area covered by it on the different dates. Subsequently, simple linear regressions at pixel level were established between the classes of LULC (represented by NDVI values) and LST. This relationship took into account the LULC images (obtained on 29 September 1989, 25 October 1999, 3 October 2008, and 16 November 2018) and corresponding LST.

In order to verify the surface temperature evolution over the historical series, the data were divided by season. Thus, under similar seasonal conditions, the data were grouped and compared using descriptive statistical analysis performed for the variables mean and standard deviation. From the surface temperatures obtained in each of the 117 images, the variables per year and respective season were calculated. It is noteworthy that there was an absence of images in some seasons, mainly due to the presence of cloudiness and/or technical failure in data collection by the sensor.

## 3. Results

The main changes in LULC during the analyzed period can be observed in the area occupied by the urban spot (Figure 3). Table 2 shows the LST and area values for each class by year of study. In 1989 the total area corresponding to urban settlements was approximately 165 hectares, that is, approximately 31% of the study area, increasing to 91% in 2018 (486.54 hectares). It is worth mentioning that the most expressive variations of this class in the period from 2008 to 2018, where its cover increased by 25.4%, compared to the 21.5% were verified between 1999 and 2008 and 13.1% between 1989 and 1999.

An adverse situation can be observed on the LULC evolution in areas classified as forest and grassland. In 1989, the areas with forest cover occupied 14.44% of the area, and in 2018, they changed to only 0.6% just as the grassland areas decreased from approximately 51% to 7.8% in the same period. This result was probably associated with the expansion process of the urban area, which, as shown in Figure 3, occurs over areas previously covered by native forest, areas in reforestation, and grassland.

The areas represented by the exposed soil class showed an oscillation in their occupation between the years 1989 and 2018. In 1989, these covered an approximate area of 19 ha, changing to 2.67 ha in 2018, that is, a decrease from 3.6% to 0.5% of the total area. However, between 1999 and 2008, there was an increase from 0.2% to 8.9%, which was probably associated with the increase in the areas of soil preparation for planting temporary crops since not all areas were urbanized and/or soil exposure due to the occurrence of deforestation since the forest areas had the greatest reduction in this period among the analyzed intervals (reduction from 13.3% to 5.6% of the total area).

In turn, the LST mean values (Table 2) obtained in each of the LULC classes, for the four years covered, were ranked as follows: Exposed Soil > Urban area > Grassland > Forest. The covers classified as Exposed Soil and Urban area qualified for intense human activity and obtained LST mean values superior to the Grassland and Forest covers. In the four classified images (Figure 4), minimum and maximum LST mean differences of 3.7 °C in 1989 and 4.5 °C in 2008 were observed between covers classified as forest and urban area. An even greater difference can be observed between the exposed soil and forest classes, in which the smallest difference reached 3.6 °C in 2008 and 5.6 °C in 2018. Therefore, measures for planning the land use and cover and cooling the surface should consider the implementation of green areas in environments where civil construction and soil exposure predominate.

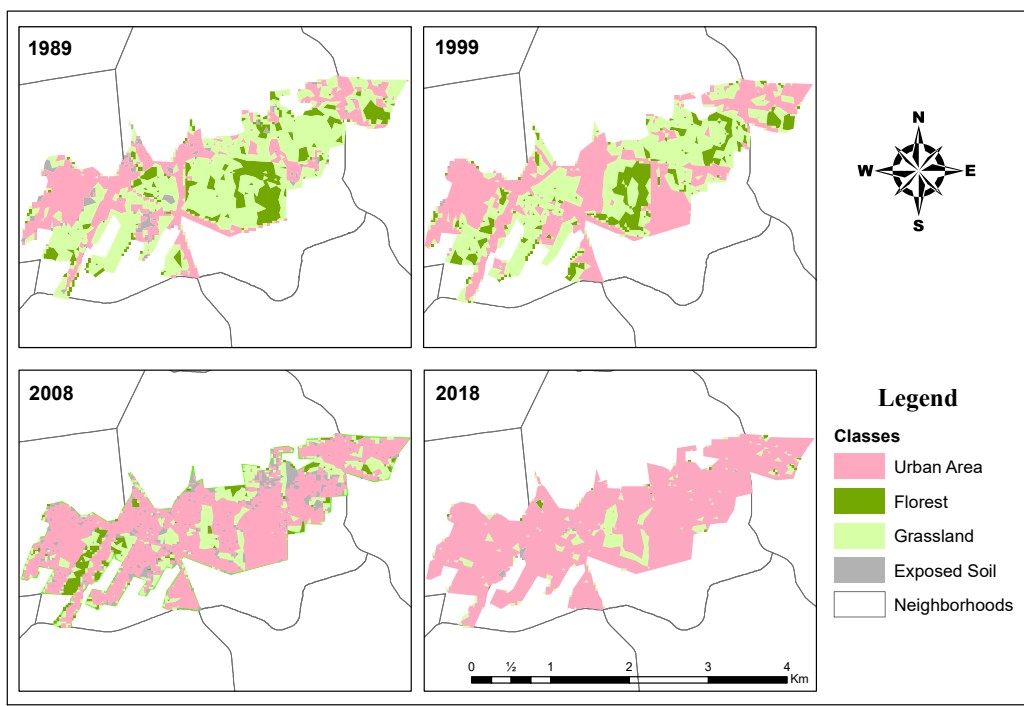

**Figure 3.** LULC maps of 1989, 1999, 2008, and 2018 of the study area.

**Table 2.** LST (°C) and area (%) means obtained in the 1989, 1999, 2008, and 2018 images from the LULC types.

| Years | Urban Area | Grassland | Forest | Exposed Soil |
|---|---|---|---|---|
| | LST (°C)/Area (%) | LST (°C)/Area (%) | LST (°C)/Area (%) | LST (°C)/Area (%) |
| 1989 | 22.5/31.0 | 20.8/51.0 | 18.9/14.4 | 22.6/03.6 |
| 1999 | 30.9/44.1 | 28.3/42.4 | 26.7/13.3 | 31.5/00.2 |
| 2008 | 29.2/65.6 | 26.9/19.9 | 24.7/05.6 | 28.3/08.9 |
| 2018 | 36.5/91.0 | 33.9/07.8 | 32.5/00.6 | 38.1/00.5 |

LST mean values obtained in areas classified as exposed soil qualified the most heated surfaces in the years 1989, 1999, and 2018 (Table 2). This result is attributed to the high thermal amplitude in short periods seen in this type of LULC, which favors the sudden increase in temperature in the face of prolonged exposure to sunlight. Thus, as a consequence, there is an intensification of the heat irradiation process for the environment, mainly on a local scale.

The annual graphic behavior of the average LST values of each LULC type associated with area variations, as well as the average decennial values presented in Table 2, pointed

to the temperature growth in all classes, as shown in the graphs in Figure 5. The annual correlation between LST values and areas of each LULC class resulted in R = 0.36 for urbanized surfaces, R = −0.34 for grassland covers, R = −0.22 for forest areas, and 0.21 for exposed soil surfaces. However, seasonally, there were better correlations between the variables for the Urban Area and Grassland classes (R = 0.49 and R = −0.50, respectively) in the autumn season and Exposed Soil (R = 0.51) in the summer, as shown in Table 3. It is associated with high annual amplitudes in the areas of Forest and Exposed Soil classes, probably associated with silviculture and soil management practices for planting annual crops, respectively.

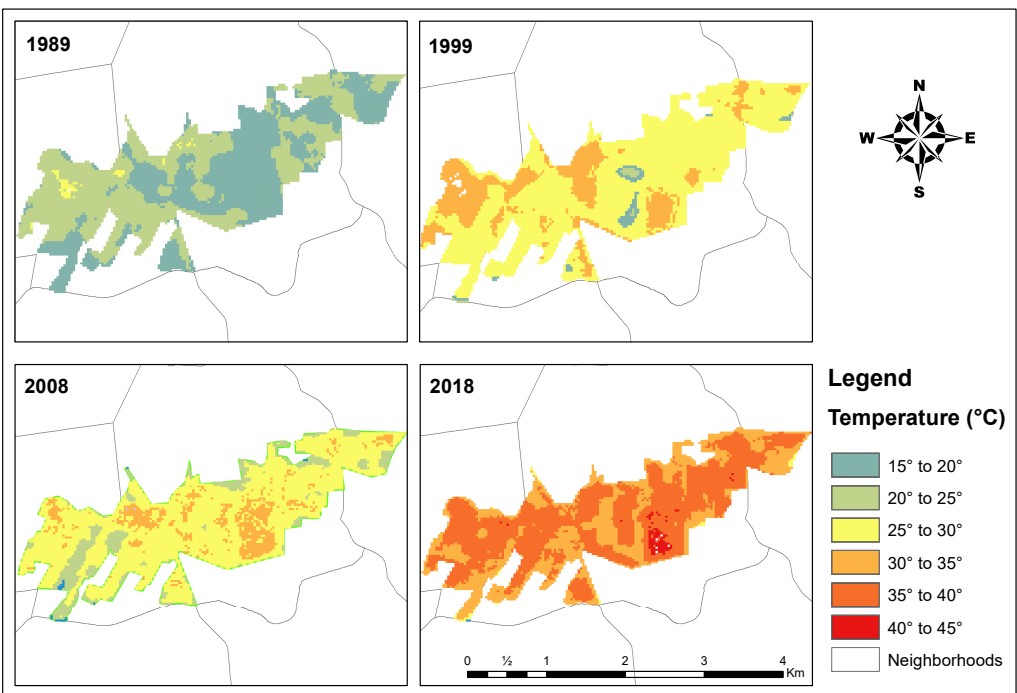

**Figure 4.** LST maps for September 1989, October 1999, October 2008, and November 2018.

The seasonal analysis of the time series shown in Figure 6 allows verifying the increase in LST of the study area in all seasons. This result is justified by the increasing average values obtained over the decennial intervals classified in Table 4. The greater amplitude between the average values of the first and last decade was verified in the autumn season. In this season, the difference between the mean temperatures verified between the first (mean of 20.20 °C) and the last decade (mean of 26.56 °C) of the analyzed period was 6.36 °C. Between the same periods, this difference was 4.09 °C in summer, 4.40 °C in spring, and 2.41 °C in winter. Probably, this difference observed in the autumn is associated with the amplitude of the LST values verified over the years, since the standard deviation resulted in the highest value among the other seasons, reaching 7.75 °C. It is worth mentioning the high amplitude of these average annual values (Figure 6) due to the low availability and even the absence of orbital images with good atmospheric conditions in some intervals of the historical series.

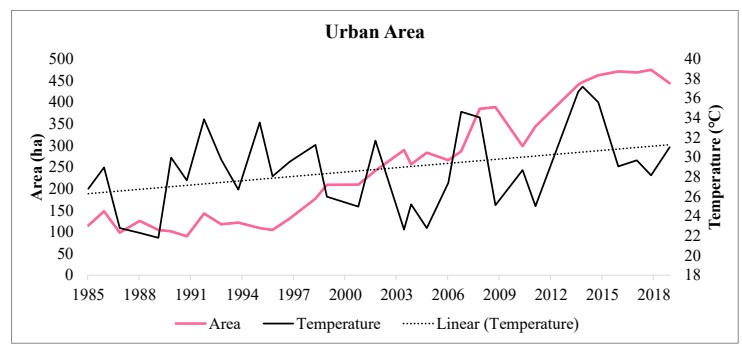

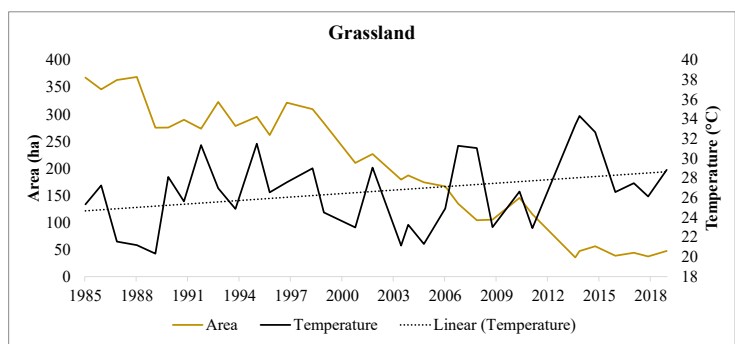

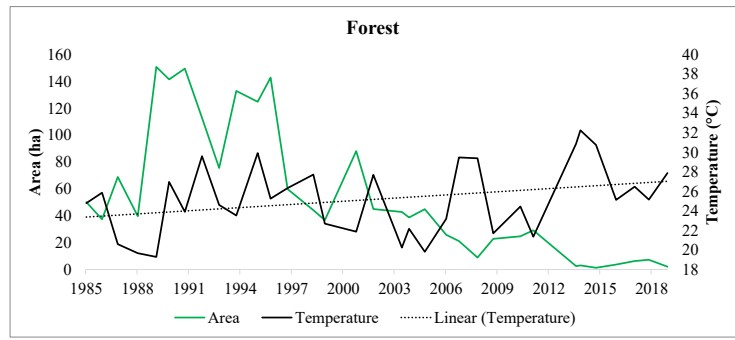

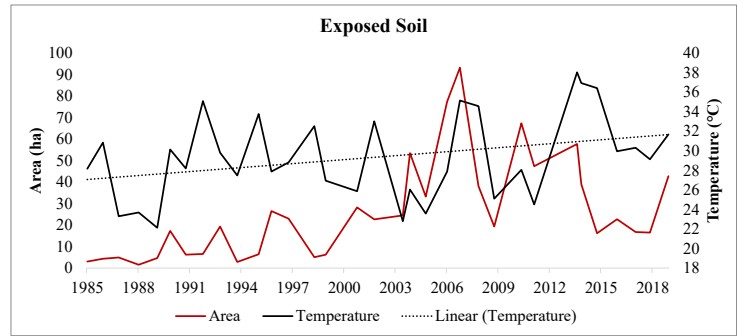

**Figure 5.** LST annual means for each land use and land cover areas in the period of 1985 to 2019.

**Table 3.** Correlation Coefficient (R) between LST sazonal means and respective areas for each LULC type.

|  | Urban Area | Grassland | Forest | Exposed Soil |
|---|---|---|---|---|
| Summer | 0.26 | −0.35 | −0.13 | 0.51 |
| Autumn | 0.49 | −0.50 | −0.26 | 0.12 |
| Winter | 0.25 | −0.16 | −0.32 | −0.06 |
| Spring | 0.25 | −0.20 | −0.16 | −0.04 |

The greatest reduction in the mean LST was observed in the winter season between the periods 1985–1999 and 2000–2008. In addition to the biggest difference observed in autumn, associated with the high standard deviation value (7.75—Table 4), this reduction in winter also resulted from the high amplitude of LST values in the 2000–2008 period (7.50 standard deviation). Thus, there are uncertainties as to whether cooling actually occurred in this season in the period of 1985–2008.

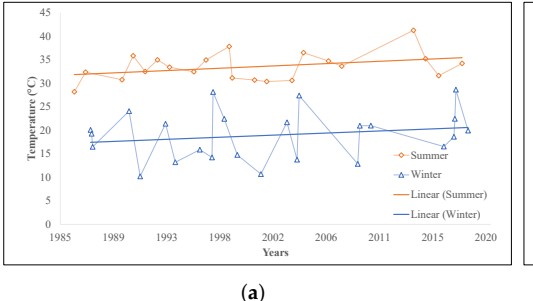
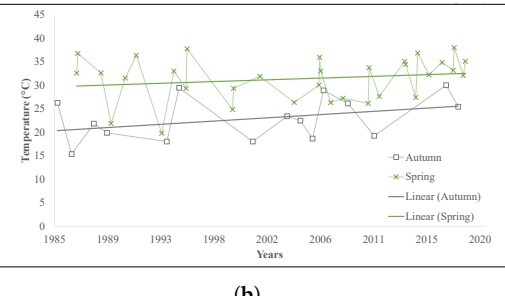

(**a**)         (**b**)

**Figure 6.** Land surface temperature means by year in the period from 1985 to 2019 of the study area. (**a**) Annual LST means for Summer and Winter. (**b**) Annual LST means for Autumn and Spring.

A trend analysis was carried out using the Mann–Kendall non-parametric test [67] at a 95% confidence interval, as shown in Table 5. The first scenario is to test the null hypothesis ($H_0$)—there is no trend in the series against the alternative hypothesis ($H_a$), and there is a trend in the series. If the computed *p*-value is lower than the significance level alpha = 0.05, one should reject null hypothesis $H_0$ and accept alternative hypothesis $H_a$. If the computed *p*-value is greater than the significance level of alpha = 0.05, one cannot reject null hypothesis $H_0$. Thus, in autumn and spring, the LST follows an upward trend across the years, when the *p*-value was lower (0.02 and 0.03, respectively) than the significance level. On the other hand, in summer and winter, no trend was found since the *p*-values were above the significance level, 0.08 and 0.38, respectively.

**Table 4.** Mean and STD (° C) of the seasons in the 1985–1999, 2000–2008, and 2009–2018 intervals.

| Intervals | Summer | | Autmumn | | Winter | | Spring | |
|---|---|---|---|---|---|---|---|---|
| | **Mean** | **STD** | **Mean** | **STD** | **Mean** | **STD** | **Mean** | **STD** |
| 1989–1999 | 32.85 | 3.07 | 20.20 | 5.85 | 18.38 | 4.83 | 29.88 | 5.70 |
| 1999–2008 | 32.81 | 3.06 | 22.76 | 7.75 | 16.95 | 7.50 | 31.14 | 3.57 |
| 2008–2018 | 36.94 | 4.14 | 26.56 | 3.88 | 20.79 | 4.69 | 34.28 | 3.84 |

**Table 5.** Trend Analysis using the Mann–Kendall test.

| | Summer | Autumn | Winter | Spring |
|---|---|---|---|---|
| Observations | 29 | 27 | 26 | 33 |
| Minimum | 27.85 | 12.64 | 9.74 | 18.88 |
| Maximum | 42.66 | 32.20 | 29.50 | 39.42 |
| Mean | 33.40 | 22.79 | 18.84 | 31.92 |
| Std. deviation | 3.41 | 5.92 | 5.33 | 4.84 |
| Kendall's tau | 0.23 | 0.32 | 0.13 | 0.27 |
| S | 94 | 111 | 41 | 144 |
| *p*-value | 0.08 | 0.02 | 0.38 | 0.03 |

It was verified a negative linear regression between NDVI and LST (Table 6). In this manner, the reduction in surface temperature is confirmed by the increase in NDVI both in urbanized areas and in their surroundings occupied by rural activities demonstrated by soil exposure and areas covered by forest and grassland. Therefore, the implementation of green spaces in areas of intense human activity contributes to improving the thermal

comfort of the population on a local scale. The low values of correlation coefficient found in the grassland and forest classes in 1989 may be associated with the influence of local meteorological conditions and forest in 2008 and 2018 and exposed soil classes in 2018, justified by the reduced number of samples due to the reduction in their areas, as shown in Table 2. In addition, smaller areas may have greater influence from the surroundings, either by the action of winds, humidity, air temperature, and by the spectral mixture.

**Table 6.** Results of Correlation Coefficient (R) obtained from the LULC classes in the years 1989, 1999, 2008, and 2018.

| Year | Urban Area | Grassland | Forest | Exposed Soil |
|------|-----------|-----------|--------|--------------|
| 1989 | −0.55 | −0.03 | 0.12 | −0.40 |
| 1999 | −0.58 | −0.30 | −0.30 | −0.72 |
| 2008 | −0.59 | −0.39 | −0.07 | −0.56 |
| 2018 | −0.76 | −0.28 | −0.16 | −0.06 |

In particular, demonstrated by the higher values of the Correlation Coefficient (R), the cover designated by the urban area quantified greater dependence on its LST compared to variations in NDVI when compared to the other covers in 1989, 1999, 2008, and 2018. In addition, there was an increase in this dependence over the study period and since in 1989, approximately 55% of the LST values were explained by the NDVI variation, whereas in 2018, this ratio increased to 76%. This result is probably attributed to the high decrease in vegetation covers (forest and grassland), as demonstrated in Table 2, and its cooling effect on the surrounding areas over the approximate 34 years period. This effect was already related by Marzban et al. [29] in a study where all LULC classes depict an inverse correlation between LST and NDVI.

The relationship between the variables can also be observed over the area characterized by soil exposure in the years 1999 and 2008 when NDVI explained 72% and 56% of the LST values, respectively. This result was probably associated with the significant increase in urban areas occurring in 1989 to 2008 (34.6% increase), given the similarity between the two types LULC mean temperatures and surrounding effects. In this manner, it is possible to affirm the relationship between the two classes in the years 1999 and 2008 justified by the correlation coefficient. The not-significant correlation value verified for this LULC type in 2018 results from the very low area classified; thus, the low number of pixels in the date is exposed in Figure 3. Furthermore, such a relationship occurred mainly in areas where the highest values of LST were recorded (anthropized surfaces).

Regarding the direct relationship between NDVI and LST in the aforementioned four years, regardless of LULC, in general, satisfactory R values were obtained. As shown in Table 7, in 1989, NDVI explained 64% of LST values, 79% in 1999, 79% in 2008, and 78% in 2018. The lowest R value verified in 1989 was associated with the predominance of grassland areas (Table 2) for which its correlation coefficient between NDVI and LST was close to zero (Table 7). Thus, probably the highest R values obtained in the following years were associated with the predominance of urbanized areas, where NDVI explained LST variations better.

**Table 7.** Results of correlation coefficient (R) obtained from the LULC classes in the years 1989, 1999, 2008, and 2018.

| Statistic | 1989 | 1999 | 2008 | 2018 |
|-----------|------|------|------|------|
| Mean LST (°C) | 21.10 | 29.25 | 28.43 | 36.42 |
| STD LST (°C) | 2.02 | 2.27 | 2.18 | 2.29 |
| Mean (NDVI) | 0.59 | 0.57 | 0.40 | 0.40 |
| STD (NDVI) | 0.15 | 0.21 | 0.19 | 0.18 |
| R (LST-NDVI) | −0.64 | −0.79 | −0.79 | −0.78 |

## 4. Discussion

In the present study, visible, near infrared, and thermal infrared images from the TM-Landsat 5 and OLI/TIRS-Landsat 8 sensors were used to classify LULC and to calculate NDVI and LST. RTE inversion was used to retrieve the LST of images collected by the sensors throughout the historic series (1985–2019), and LULC classification was performed in the spring of the years 1989, 1999, 2008, and 2018 in order to verify the influence of LULC classes on the temperature from NDVI on a temporal and spatial scale.

Seasonally, the highest correlations between LST and area size were verified on the Urban Area and Grassland classes classified in the autumn season and Exposed Soil in the summer. The surfaces classified as Urban Area, Grassland, and Exposed Soil had approximately 50% of the LST values explained as a function of the variation in the occupied area over the period covered. In turn, in areas classified as Forest, this correlation was relatively weak (32% in winter). Zhou et al. [68] verified an adverse situation in a study carried out in the northeast of the USA, state of Maryland, where the percentage of imperviousness explained approximately 50% of the total variation in LST in the wintertime and up to 77.9% during summer.

This study also analyzed long-term seasonal trends in LST during the last three decades. The calculation of the average decadal differences of LST allowed verifying the increases for all seasons, which were more expressive in autumn (6.36 °C), intermediate in spring and summer (4.40 °C and 4.09 °C, respectively), and less expressive in winter (2.41 °C). However, for annual averages, by using the Mann–Kendall test, trends of temperature increase were observed only in the autumn and spring seasons. In a similar study carried out in Spain, Khorchani et al. [69] verified the Mann–Kendall trend of temperature increases in the four seasons of the year, which were more expressive in summer, intermediate in spring and autumn, and less expressive in winter. In addition, the authors attributed the increase in summer temperature to the strong increase in Summer sunshine duration (SUN) in Spain between 1982 and 2014. Thus, for future research, we recommend the analysis of this variable together with the trends of increase or decrease in LST.

The highest values of the correlation coefficient, between the LST and NDVI variables, were calculated in the LULC classes designated as Urban Area and Exposed Land. This result points to the dependence of LST values on the presence of vegetation in areas where anthropogenic interference occurred exactly, either through soil impermeability or the removal of vegetation cover exposing the surface. Thus, human actions qualified the increase in temperature in these areas in relation to neighboring environments, quantifying the phenomenon of Urban Heat Islands (UHIs). Similar results were verified in Yue et al. [70] and Hereher [71].

The increase in temperature observed for each season from 1985 to 2018 is in line with the scenario projected by the IPCC (Intergovernmental Panel on Climate Change, Geneva, Switzerland) in its Fifth Assessment Report (AR5) [72]. According to the report, at the end of the century, the increase in mean temperature should be 2.6 to 4.8 °C, which is mainly attributed to the growth of cities and their emission of 40% of the gases responsible for potentiating the greenhouse effect [73,74].

In a study carried out on the municipality of Porto Alegre- Rio Grande do Sul, Brazil, Grondona et al. [75] verified an increase in the average LST during the period from 1985 to 2005 with a 4.13% reduction in the vegetated area and an increase of 24.22% in the urbanized area. The present study found that the removal of vegetation cover and the impacts caused on the LST are of great complexity. Smaller correlation coefficients (from 0.12 to −0.39), between LST and NDVI, were found in vegetated areas (field and forest) and higher (from −0.55 to −0.76) in urbanized areas. It is important to note that, in terms of remote sensing, vegetated areas present greater homogeneity (lower spectral variability and types of materials) when compared to urbanized areas (greater geometric and spectral variability, especially of the materials used in construction). Such a result is probably associated with local meteorological variables (rainfall, air humidity, and wind) and physicochemical factors inherent to these roofs.

Although many factors (meteorological, topography, and roughness) are responsible for LST variations, the extension and arrangement of the areas are fundamental for understanding the relationship of this variable with vegetation. The results of this study demonstrate the temporal and spatial evolution of the interaction between vegetation and thermal dynamics through linear regression analysis at the pixel level. For urban areas, the correlation strength between LST and NDVI increased with time. In the study carried out in Shanghai, China, Weng et al. [35] concluded that when urban structures and built-up areas occupy most of a study area, their thermal surface will become spatially homogenized. Thus, we conclude that the highest values of R (0.72 in 2018) between LST and NDVI may be associated with the expansion of the urban area (195% increase for 1989 until 2018 period), causing changes in the energy balance and changes in local weather conditions, which are responsible for configuring an environment urban area with homogeneous thermal conditions.

Since several studies report high correlation coefficients (above 0.9) between NDVI and LST for urbanized areas that were already large at the beginning of the historical series, as shown in Baghdad, Iraq with 227,800 ha in 1984 [76], Adama Zuria District-Ethiopia with 64,800 ha in 1989 [77], and Dhaka metropolitan area-Bangladesh with 802,200 ha in 2000 [78], we observed the smaller correlation value (−0.55) obtained where the small urban nucleus occupied only 165 ha in 1989. Thus, for urban planners and urban policymakers, it is necessary to understand that the influence of vegetation presence to mitigate the increase in temperature is dependent on the spatial arrangement and coverage area of urbanized surfaces. The urban thermal environment planning of new neighborhoods, condominiums, and industrial parks must take into account not only the presence of vegetation but also biophysical, meteorological, and topographic parameters, and other LULC indices (e.g., Normalized difference built-up index, Enhanced vegetation index, Soil adjusted vegetation index, Modified normalized difference water index, Normalized difference mud index, etc.) may be examined to find a better correlation with LST.

LST is considerably influenced by vegetation dynamics [20,79]. NDVI is widely used to assess changes in LST [14,79,80]. Therefore, we chose to use it in this work. NDVI can suppress a significant amount of the noise caused by atmospheric effects, clouds or cloud shadow, topography, and changing sun angles [31,81]. However, it is worth noting that it is sensitive to canopy background variations and more saturated at high biomass levels [31,82–84]. The Enhanced Vegetation Index (EVI), with a more dynamic range, can be considered as an improvement of the NDVI, concerning the saturation in dense forest, soil reflectance influence, and atmospheric correction [31,85]. Some papers have analyzed the relationship between LST and EVI [31,79,84,86,87] and so we intend to include this index in future research.

## 5. Conclusions

The main conclusions were as follows: (1) The LST retrieved by the RTE inversion of Landsat 5 and 8 data showed consistency both for atmospheric parameters obtained from MODTRAN (1985–1999) and for NASA's online calculator (2000–2018). (2) Throughout the period from 1989 to 2018, the area where the current urban spot belonging to the Hípica, Aberta dos Morros, and Restinga neighborhoods has undergone significant changes in its LULC. The forest and grassland classes showed a decrease of approximately 98% and 79% in their areas, respectively, while the classes represented by the urban spot and surface exposure areas suffered an increase of 31% and 3.5% in 1989 to 75% and 14% in 2018 of the study area, respectively. (3) From annual mean values of temperature, the Mann–Kendall test allowed observing the trend of temperature growth in the autumn and spring seasons. In autumn, LST growth was more expressive, and its difference reached 6.36 °C when compared to the first and last decades of the historical series. In spring, this difference was 4.4 °C. (4) Multiple comparative analyses indicated that the difference in LST between most types of LULC was significant, with areas of exposed soil with the highest LST (mean of the study area 38.1 °C) and forest land with the lowest LST (mean of 18.9 °C), which may

indicate the formation of urban heat islands on the study area. (5) The study allowed the demonstration of the influence of the urbanization process and the removal of vegetation cover on the significant increase in temperature. High surface temperatures were associated with areas of high urban concentration for both the images classified in 1989 and 2018, but an increase in a mean of 4.18 °C was observed when compared to the first and last decade of the historical series in the four seasons. This conclusion is attributed to the significant expansion of the urban spot and the areas of exposed soil over the forest and grassland areas. Thus, the interference of the characteristics of the materials covering the surface over the heat of the environment where they are located is evident. (6) The relationship between NDVI and LST verified for the urban area, exposed soil, grassland, and forest classes showed a negative linear correlation, as expected. A higher correlation coefficient (R) was obtained between NDVI and LST in the areas classified as urbanized area and exposed soil, where the highest temperatures were found. (7) The relationship between NDVI and LST in urbanized areas showed an increase associated with the correlation coefficient from $-0.55$ in 1989 to $-0.76$ in 2018. This result is probably related to the growth of the urban area (from 31% of the area in 1989 to 91% in 2018), causing surface homogeneity (lesser meteorological variations, reduced effect of the surrounding areas, and less spectral mixing). A different behavior was observed in other areas (field, forests, and exposed soil), where R did not present any pattern of evolution.

Some drawbacks to this work should not be overlooked. First, the meteorological variables, land surface albedo, [88] and landscape metrics [89] can provide greater accuracy to the results and discussions, since these variables directly influence LST. Second, the low availability of satellite images, especially during the winter when clear sky conditions are less frequent, render the trend analysis of increases or decreases in LST over the period difficult. Future studies should make use of complementary products to meet this demand, for example, the Aster sensor LST product. Finally, the results obtained were generated from high-resolution remote sensing images and geoprocessing tools; however, the LST was not validated with field measurements.

The information gathered about the surface temperature, land use, and cover using remote sensing techniques, using visible and thermal infrared bands from the Landsat 5 and 8 satellites, TM, and OLI sensors, respectively, allowed the visualization and understanding of the microclimate dynamics of an urban area in the city of Porto Alegre, RS-Brazil, over the period studied, as well as its relationship with the different types of surface cover. The historical characterization of the relationship between LST and NDVI according to the types of LULC made it possible to quantify the impacts of the rapid urbanization process in a small urban nucleus as a model for future decision making regarding urban expansion and planning. Thus, the study highlights the importance of air humidity and evapotranspiration processes, which should be maintained through the adoption of policies that mitigate the effects caused by urban expansion and densification.

**Author Contributions:** Conceptualization, E.A.K., S.B.A.R. and A.E.B.G.; methodology, E.A.K., A.E.B.G. and C.L.H.; software, E.A.K.; investigation, E.A.K., S.B.A.R., A.E.B.G., C.L.H., R.d.M.L., P.S.K., N.S.d.R. and L.R.D.; writing—original draft preparation, E.A.K., S.B.A.R., A.E.B.G. and C.L.H.; writing—review and editing, E.A.K., S.B.A.R., A.E.B.G., C.L.H., P.S.K., N.S.d.R. and L.R.D.; supervision, S.B.A.R., A.E.B.G., C.L.H. and R.d.M.L.; project administration, S.B.A.R. All authors have read and agreed to the published version of the manuscript.

**Funding:** This research received support from Coordenação de Aperfeiçoamento de Pessoal de Nível Superior—Brazil (CAPES).

**Institutional Review Board Statement:** Not applicable.

**Informed Consent Statement:** Not applicable.

**Data Availability Statement:** Not applicable.

**Acknowledgments:** The authors would like to thank the Coordenação de Aperfeiçoamento de Pessoal de Nível Superior—Brazil (CAPES) and the Laboratory of Geological Remote Sensing (Lab-SRGeo/UFRGS) for supporting this study.

**Conflicts of Interest:** The authors declare no conflicts of interest.

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
