# Peer review of "Spatiotemporal Influences of LULC Changes on Land Surface Temperature in Rapid Urbanization Area by Using Landsat-TM and TIRS Images"

_atmosphere, doi:10.3390/atmos13030460_

Round 1

Reviewer 1 Report

The manuscript presents a spatiotemporal influences of LULC changes on LST in rapid urbanization area through different images. Although not much troublesome, the abstract is a bit short and the authors could consider increasing it by some important results. Also, it would be nice if you merge paragraphs in introduction, and add some recent literature reviews about the LULC and urbanisation. Some minor comments: - line 104: Add information about yearly evatranspiration of the study area. - Figure 6: Trend-lines can be test with some statistical analysis (Sen-slope or...). - Add at least two limitation of your study in Conclusion.

Reviewer 2 Report

Comments from the editors and reviewers:

Reviewer#1

An interesting case study by the authors. The premise of the study seems weak – it is nobody’s case that the study measures changes in Land Surface Temperature (LST) caused by the urbanization process in an area previously covered by vegetation. Nonetheless, the data is interesting and quantify the impacts 12 of the rapid urbanization process in a small urban nucleus. Some additional comments are provided below for the authors to address.

Abstract:

Do the study city refer to size by area or size by population?

Line-14 need one line policy implication

Line-15: Keyword is not completed, authors should add the study area name

There are several grammatical errors all over the manuscript. English quality is quite poor. Please proofread.

Introduction:

Line 78-80: This hypothesis seems self-evident. Motivation and literature section is slightly handled. Author needed to improve it substantially. How this work could enhance current knowledge base is certainly missing, especially in previous studies have focused on the consequences of the impact of urbanization on urban thermal environment of Bangladesh region. Following work could be useful to develop motivation of the work;

doi: 10.3390/cli10010003

Material and methods

Material and methods are hard to understand because the research steps are not adequately described. I would recommend the introduction of an explicit phased methodological diagram.

The reason for the authors to select the research data, i.e. during the period between 1985 and 2018?

What is the justification of the study city for this research?

There are many types of vegetation index. Why did you choose the NDVI for this study? You actually used NDVI but there are also additional vegetation ratios presenting better performance (e.g. EVI, etc.). Please provide a paragraph describing the problems related to NDVI ratio for vegetation density estimation and mention that in future research efforts you might consider EVI, etc.

Discussion

Scholarly discussion is missing.  Authors should highlight more clearly how the analysis can support the implementation of the measures proposed by the policy making of the study areas. Moreover, Discussions need to be more scientific. It seems necessary to compare the results of this study with those of previous studies. As they are they highlight the limitations of the research.

Conclusion:

poorly written, use above works to cross check your findings. Your work and suggested works may differ in terms of area being worked, focus on main novelty, future research direction, however you should find a way to deal with this

Reviewer 3 Report

The changes in Land Surface Temperature (LST) caused by the urbanization process and the relationsphis with the LSD increase is surely a current research topic.

The work is well written and the methodology reported does not need additional descriptions. I would specify the resolution of the LST maps (30m?). The results show some effects and when the correlation coefficients are found to be low (no correlation) the explanation given I believe is justified. In this case, however, I would not draw conclusions on whether or not to increase the LST as a function of the NDVI in those cases as the relationships are not robust. I would therefore review this part where, however, a justification has already been given.

Figure 5 : it is shifed

Round 2

Reviewer 2 Report

Accept

Author Response

Dear Reviewer 2,

Thank you in advance for your revision. 

They were essential for the article improvement.

Kind regards,

Authors